# Dimensional Stability of Treated Sengon Wood by Nano-Silica of Betung Bamboo Leaves

Istie Rahayu [1,*], Fitria Cita Dirna [1], Akhiruddin Maddu [2], Wayan Darmawan [1], Dodi Nandika [1] and Esti Prihatini [1]





[1] Department of Forest Products, Faculty of Forestry and Environment, IPB University, Bogor 16680, Indonesia; fitriacita@gmail.com (F.C.D.); wayandar@indo.net.id (W.D.); nandikadodi@gmail.com (D.N.); esti@apps.ipb.ac.id (E.P.)
[2] Department of Physics, Faculty of Mathematics and Natural Sciences, IPB University, Bogor 16680, Indonesia; maddu3@gmail.com
[*] Correspondence: istiesr@apps.ipb.ac.id

**Abstract:** Sengon (*Falcataria moluccana* Miq.) is one of the fastest growing wood that is broadly planted in Indonesia. Sengon wood has inferior wood properties, such as a low density and dimensional stability. Therefore, sengon wood requires a method to improve its wood quality through wood modification. One type of wood modification is wood impregnation. On the other hand, Betung Bamboo leaves are considered as waste. Betung Bamboo leaves contain silica. Based on several researches, nano-$SiO_2$ could improve fast-growing wood qualities. According to its perfect solubility in water, monoethylene glycol (MEG) is used in the study. The objectives are to evaluate the impregnation treatment (MEG and nano-silica originated from betung bamboo leaves) in regard to the dimensional stability and density of 5-year-old sengon wood and to characterize the treated sengon wood. MEG, MNano-Silica 0.5%, MNano-Silica 0.75%, and MNano-Silica 1% were used as impregnation solutions. The impregnation method was started with 0.5 bar of vacuum for 60 min, followed by 2.5 bar of pressure for 120 min. The dimensional stability, density, and characterization of the samples were studied through the use of scanning electron microscopy (SEM), energy dispersive X-ray spectroscopy, X-ray diffraction (XRD), and Fourier transform infrared spectroscopy (FTIR). The results show that the treatment had a significant effect on the dimensional stability and density of sengon wood. Alterations in the morphology of treated sengon wood were observed through the full coverage of the pits on the vessel walls (SEM analysis results) and the detection of ethylene (FTIR analysis results) and silica (XRD and FTIR analysis results). Overall, the 0.75% MNano-Silica treatment was the most optimal treatment for increasing the dimensional stability and density of 5-year-old sengon wood.

**Keywords:** betung bamboo; dimensional stability; impregnation; MEG; nano-silica; sengon

## 1. Introduction

Degradation and deforestation decrease wood production from our natural forests. Therefore, we need alternative wood resources to overcome the wood supply shortage through the utilization of fast-growing species [1] or a wood biomass [2–4]. Sengon (*Falcataria moluccana* Miq.) is a type of fast-growing tree that is widely planted in community forests and community plantation forests in Indonesia. Sengon has a short harvest time. Owing to the rapid growth of the tree, the wood has a low density, strength, and durability, as well as a high portion of it being juvenile wood [5]. Based on the research results of [6], 5- to 7-year-old jabon and sengon trees contain up to100% juvenile wood. The density and hardness of sengon wood are 0.3–0.5 g/$cm^3$ and 112–122 kg/$cm^2$, respectively. Sengon wood is classified into a durability class of IV–V and strength class of IV–V [7]. Nowadays, sengon wood in Indonesia is used for pulpwood, light construction, furniture, and wood composite. Sengon wood production in Indonesia in 2019 was 5.47 million $m^3$ [8]. Therefore, the quality of

fast-growing wood needs to be improved through technology. Wood modification technology has been discussed in many journals in recent decades; however, only a few technologies have led to industrial application. One of them is thermal modification. It is a potential technology to be developed in industries. It improves the wood's hygroscopicity, dimensional stability, and durability without harming the environment [9–11]. However, heat treatment degrades mechanical properties [12,13], decreases wood durability against termites [14–16], and causes it to not be resistant to weathering [17]. In line with thermal modification, chemical modification is a promising alternative for improving wood qualities. There are two chemical modifications with the potential to be developed on an industrial scale: acetylation and furfurylation [18]. The impregnation of molecular compounds into the wood structure is also another option being able to increase the density and dimensional stability of wood. Research on impregnation technology for fast-growing wood has been carried out in Indonesia and other countries, using methyl methacrylate [19–21], formaldehyde-based chemical compounds such as phenol formaldehyde [22,23], rosin [24], furfuryl alcohol [25], paraffin [26], and aqueous solutions [27]. The utilization of chemical substances and formaldehyde material for wood modification could have a hazardous consequence to our health and environment. Therefore, we need to find other material alternatives that are more environmentally friendly.

Silica is a chemical that has wide applications in various fields, including being used as a polymer in wood impregnation. Silica can be obtained from the commercial markets or extracted from natural materials, such as bamboo leaf ash, as shown in research [28]. Bamboo leaves are considered waste [29]. Researchers analyzed bamboo leaf ash and found a silica content of 79.93%, which was the second largest content after rice husk ash (93.2%) [30]. Based on the findings of [31], nano-$SiO_2$ could effectively improve the dimensional stability and density of jabon wood. Similar outcomes were reported by [32] using sengon wood and nano-$SiO_2$ and [33] using poplar wood (Populus spp.) treated with furfuryl alcohol and nano-$SiO_2$. Therefore, in this study, we produced nano-silica created from betung bamboo leaves.

Monoethylene Glycol (MEG) is perfectly soluble in water, colorless, liquid, odorless, has a low volatility and has a 62.07 g/mol molecule weight [34,35]. According to these characteristics, MEG was used. In the current study, we aim to evaluate the impregnation treatment (MEG and nano-silica derived from betung bamboo leaves) with regards to the weight percent gain (WPG), anti-swelling efficiency (ASE), bulking effect (BE), water uptake (WU), and density of 5-year-old sengon wood, and to characterize the treated sengon wood.

## 2. Materials and Methods

This research was divided into several stages, namely, the preparation of wood samples; preparation of impregnation solutions; the impregnation process; the calculations of the dimension stability and density. In this study, the materials used were bamboo betung leaves; 5-year-old sengon wood from community forests in Sukabumi, West Java, Indonesia; monoethylene glycol (MEG); demineralized water. The analytical techniques included scanning electron microscopy (SEM), energy dispersive X-ray spectroscopy (EDX), X-ray diffraction (XRD), and Fourier transform infrared spectroscopy (FTIR). Bamboo betung leaves used in the form of nano-silica in this study [28] had a particle size of about 234.49–851.36 nm, where the average size was 472.67 nm with a PDI value (Particle Dispersion Index) of 0.0670.

### 2.1. Preparation of Wood Samples

The sample size of the sengon wood for testing was 2 cm × 2 cm × 2 cm [36]. Testing encompassed WPG, ASE, WU, BE, and oven-dried density. Each treatment had a total of 5 replicated samples.

## 2.2. Preparation of Impregnation Solutions

Impregnation solutions were prepared by mixing MEG and nano-silica by using an ultrasonic processor (amplitude of 40% for 60 min). In Table 1, it can be seen that there were several treatments for the composition of the MEG and MEG solutions and nano-silica (MNano-Silica).

**Table 1.** The volume of MEG and nano-silica as impregnation solutions.

| Treatment | MEG 50% (mL) | Nano-Silica (g) |
|---|---|---|
| MEG | 1000 | 0 |
| MNano-Silica 0.5% | 1000 | 5 |
| MNano-Silica 0.75% | 1000 | 7.5 |
| MNano-Silica 1% | 1000 | 10 |

## 2.3. The Impregnation Process

The impregnation process with MEG and the MNano-Silica solutions was carried out by adapting a previously reported method [32], in which sengon wood was impregnated using MEG and nano-$SiO_2$. Dimensional measurements and weighing were carried out to test the WPG, BE, and WU [37], ASE [38], and the oven-dried density.

## 2.4. Calculation of Dimensional Stability and Oven-Dried Density

Calculations for WPG, ASE, BE, WU, and oven-dried density were conducted using the following formulas:

$$\text{WPG (\%)} = [(W1 - W0)/W0] \times 100 \quad \text{ASE (\%)} = [(Su - St)/Su] \times 100$$

$$\text{WU (\%)} = [(W2 - W1)/W1] \times 100 \quad \text{BE (\%)} = (V1 - V0)/V0 \times 100$$

$$\rho \left(\text{g/cm}^3\right) = \frac{B}{V} \times 100$$

where:

W0 = the oven-dried weight of sample before treatment;
W1 = the oven-dried sample weight after treatment;
W2 = the weight of the sample after immersion in water for 24 h;
V0 = the oven-dried volume of sample before treatment;
V1 = the oven-dried sample volume after treatment;
Su = the volume shrinkage of untreated wood;
St = the volume shrinkage of treated wood;
B = the weight of the sample before or after treatment;
V = the volume of the sample before or after treatment.

ASE values were determined by cycled water soaking method [38]. It was repeated 3 times, while oven-dried density after treatment was evaluated by taking into account WPG and BE values and untreated oven-dried density as a baseline.

## 2.5. Data Analysis Procedure

The simple, completely randomized design with 1 factor was used in this study, the factor being, namely, the treatment variation factor at 4 levels, namely, MEG, MNano-Silica 0.5%, MNano-Silica 0.75%, and MNano-Silica 1%. Statistical analyses were carried out by using IBM SPSS Statistics software (version 25.0), and the Duncan test at $\alpha = 1\%$ was conducted if there was a significant difference.

## 3. Results

Dimensional stability testing on sengon wood included the calculation of the WPG, ASE, WU, BE, and oven-dried density. The results of the dimensional stability tests for sengon wood can be seen in Table 2.

**Table 2.** Testing the dimensional stability of sengon wood.

| Treatment | WPG (%) | ASE (%) | WU (%) | BE (%) | Oven-dried Density (g/cm$^3$) |
|---|---|---|---|---|---|
| MEG | 27.18 [a] (±3.42) | 53.28 [a] (±6.73) | 101.68 [d] (±1.91) | 2.89 [a] (±0.84) | 0.38 [a] (±0.02) |
| MNano-Silica 0.5% | 44.61 [b] (±2.29) | 72.98 [b] (±3.88) | 76.16 [c] (±3.30) | 4.85 [b] (±0.63) | 0.42 [a] (±0.15) |
| MNano-Silica 0.75% | 50.20 [c] (±1.43) | 83.36 [c] (±4.19) | 67.99 [b] (±2.81) | 9.78 [c] (±0.94) | 0.42 [a] (±0.03) |
| MNano-Silica 1% | 58.03 [d] (±2.39) | 87.69 [c] (±6.15) | 56.34 [a] (±3.88) | 10.85 [c] (±1.37) | 0.46 [b] (±0.01) |

Note: Values in parentheses indicate the standard deviations. [a–c] Values followed by the different letters show real difference according to the Duncan test.

The WPG value for sengon wood increased along with the nano-silica concentration, presumably because the MEG and nano-silica entered the sengon wood structure. The ASE value of sengon wood also increased with each treatment. These results show that the MEG and nano-silica solutions increased in concentration in each treatment. The WU value decreased as treatments used a greater concentration of nano-silica. This result was correlated with the high WPG and ASE value. The BE value of sengon wood also increased with each treatment using a high nano-silica concentration, indicating that the added MEG and nano-silica functioned as bulking agents. The oven-dried density value of sengon wood increased with each treatment using a higher level of nano-silica. The percentage of increased oven-dried density after being treated was 26% (MEG), 40% (MNano-Silica 0.5% and 0.75%), and 53% (MNano-Silica 1%). Overall, the analysis of variance showed that each MEG and nano-silica impregnation treatment had a significant effect on sengon wood. Results were influenced by the various concentrations (0.5%, 0.75%, and 1%) of nano-silica used.

### 3.1. Characteristics of Impregnated Sengon Wood

3.1.1. SEM-EDX Analysis

Figure 1a shows pits in the vessel wall of sengon wood that were not closed. In Figure 1b, through the addition of 0.5% MNano-Silica, the sengon wood underwent morphological changes. Figure 2a shows that, with the addition of 0.75% MNano-Silica, nano-silica could enter the pits in the sengon wood, adhere to the vessel walls, and almost entirely cover all of them. A similar result was found with the addition of MNano-Silica 1% (Figure 2b), with nano-silica covering the entire surface. Additional information obtained from the analysis by SEM included EDX data, which are described in Table 3.

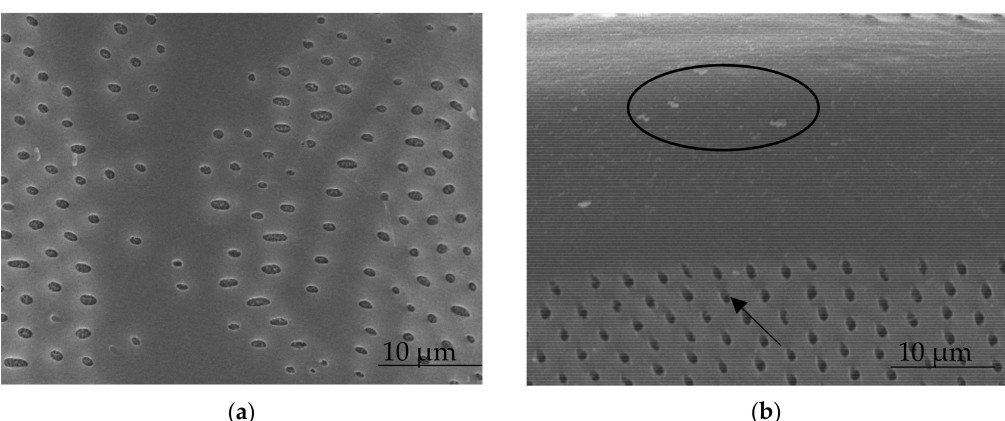

|(a)|(b)|

**Figure 1.** Morphology of wood (**a**) sengon MEG and (**b**) sengon MNano-Silica 0.5%.

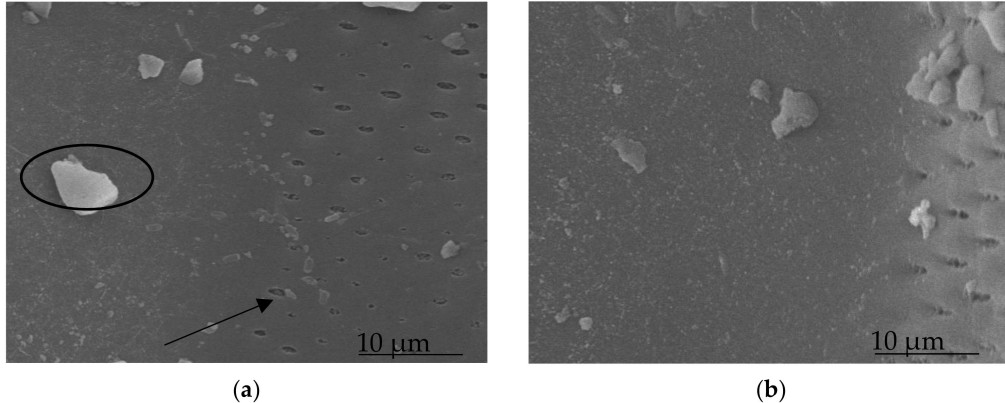

(**a**)                                  (**b**)

**Figure 2.** Morphology of wood (**a**) sengon MNano-Silica 0.75% and (**b**) sengon MNano-Silica 1%. Magnification 1000×.

**Table 3.** Chemical content of treated sengon wood.

| Treatment | Silicon (wt. %) |
|---|---|
| MEG | 0 |
| MNano-Silica 0.5% | 0.35 |
| MNano-Silica 0.75% | 0.59 |
| MNano-Silica 1% | 0.62 |

### 3.1.2. XRD Analysis

In sengon wood (Figure 3), MEG treatment was detected as high-intensity cellulose peaks with an angle of 2θ = 22.47°. The silica peaks were detected with a high-intensity treatment of MNano-Silica 0.5%, MNano-Silica 0.75%, and MNano-Silica 1% at angles of 2θ = 20.59°, 20.51°, and 21.06°, respectively, which were close to the silica peak in the JCPDS 44-1394 database, namely, angles of 2θ = 20.54° and 21.13°. In the MNano-Silica 0.5%, MNano-Silica 0.75%, and MNano-Silica 1% treatments, cellulose peaks were detected at angles of 2θ = 22.61°, 22.67°, and 22.19°, respectively, which corresponded to the cellulose peaks in the JCPDS 03-0226 database, namely an angle of 2θ = 22.26°. Furthermore, cellulose peaks were also detected in the MEG treatment for an angle of 2θ = 17.04°. This result was close to the peak of cellulose in the JCPDS 03-0226 database, namely, an angle of 2θ = 17.13°.

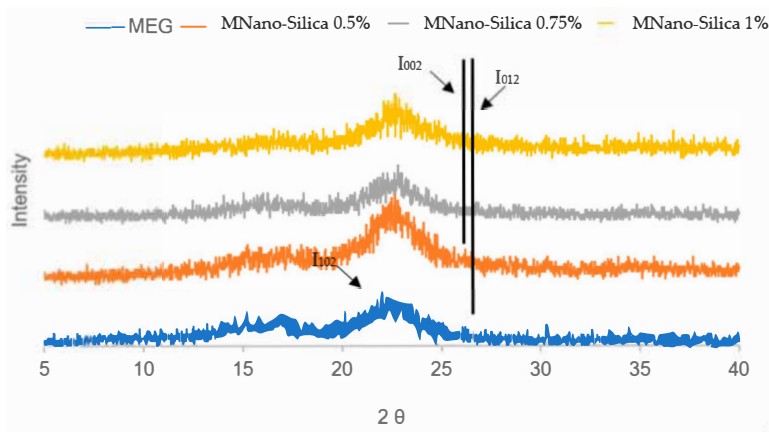

**Figure 3.** XRD spectra of sengon wood after impregnation treatment.

In the MNano-Silica 0.5% treatment of sengon wood, a silica peak of 2θ = 18.61° was detected, which was in accordance with the JCPDS 44-1394 database, namely, an angle of

$2\theta = 18.54°$. Furthermore, in the MEG treatment of sengon wood, cellulose peaks were detected for an angle of $2\theta = 17.04°$. These result were close to the peak of cellulose to the JCPDS 03-0226 database, namely, an angle of $2\theta = 17.13°$. This peak disappeared with the addition of nano-silica used in the treatment. Additional information obtained during XRD testing included data on the degree of crystallinity, which can be seen in Table 4. The results showed that the crystallinity degree of the sengon wood, augmented with the MEG treatment compared to the MNano-Silica 0.75% treatment, indicated that the addition of nano-silica could increase crystallinity.

**Table 4.** Crystallinity degree of treated sengon wood.

| Treatment | Crystallinity Degree (%) |
|---|---|
| MEG | 24.00 |
| MNano-Silica 0.5% | 27.94 |
| MNano-Silica 0.75% | 30.79 |
| MNano-Silica 1% | 29.22 |

The crystallinity degree in sengon wood decreased with the addition of 1% (29.22%) nano-silica, presumably because the resulting silica had an amorphous structure and the resulting nano-silica had a semi-crystalline structure. The nano-silica produced was semi-crystalline, which was likely influenced by the sonication treatment used.

### 3.1.3. FTIR Analysis

Figure 4 shows the FTIR spectrum of five sengon wood samples with a wavenumber ranging from 500 to 4000 cm$^{-1}$. The peak of the Si–O–Si vibrational silica was detected at the wavenumber 1060.45 cm$^{-1}$ with the addition of MNano-Silica 0.5% and strengthened by the appearance of wave numbers 1056.39 cm$^{-1}$ and 1112.19 cm$^{-1}$ with the addition of MNano-Silica 0.75%. Very similar results were also shown for the addition of MNano-Silica 1%, with wavenumbers 1058.11 cm$^{-1}$ and 1113.15 cm$^{-1}$ (Figure 4).

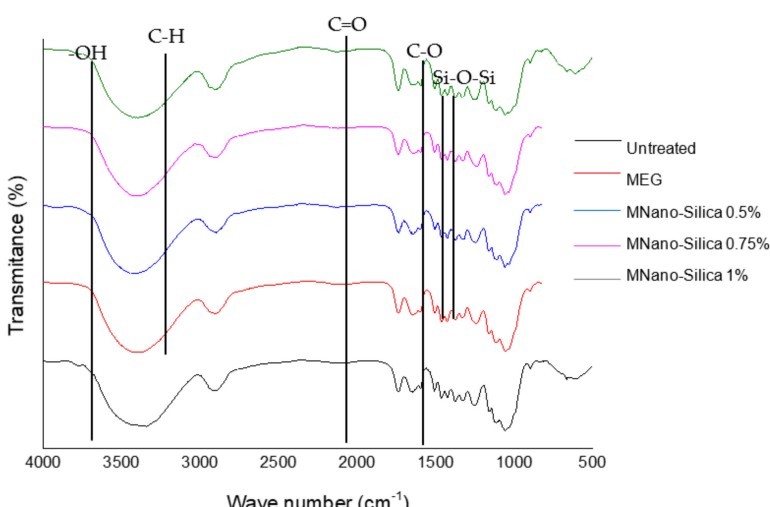

**Figure 4.** FTIR analysis of sengon wood after impregnation treatment.

### 4. Discussion

The impregnation using a polymer caused it to penetrate the wood, which could be followed by bonding with the constituent components of cell walls as described in [37]. Due to the increased ASE value, the wood's ability to adsorb water was reduced to vapor from the surroundings [28]. This finding indicated that the wood's ability to absorb water became lower with each treatment. The higher the BE value, the more polymer filling the

cell walls in the wood, which could increase the dimensional stability of the wood. This outcome was in line with the increase in the WPG value from one treatment to the next. The increase in the WPG value of sengon wood was directly comparable to the BE value. The higher the WPG values, the higher the oven-dried density of sengon wood.

This result corresponded with research reported by [32], who showed that the replenishment of nano-silica in the impregnation of sengon wood had a significant effect on each treatment. For sengon wood treated by nano-silica derived from the leaves of bamboo betung, the 0.75% MNano-Silica treatment was optimal for increasing the dimensional stability and the oven-dried density of fast-growing sengon wood.

### 4.1. Characteristics of Impregnated Sengon Wood

### 4.1.1. SEM-EDX Analysis

MNano-Silica was deposited into vessel walls and covered the cell wall of sengon wood. The image generated from the SEM-EDX test showed that the MNano-Silica 0.5% had a significant effect on the morphology of sengon wood. The MNano-Silica 0.75% resulted in better morphological changes compared with a concentration of 0.5%. Overall, these results indicate that the replenishment of nano-silica at various concentrations could affect the morphology of sengon wood.

The SEM analysis above explains the cause of this treated sengon wood that underlies the decrease in the WU value and the increase in the value of the dimensional stability parameters and oven-dried density. The nano-silica addition could improve the distribution of the MNano-Silica solution. Based on the above results, the MNano-Silica 0.75% treatment showed optimal results compared with other treatments.

### 4.1.2. XRD Analysis

Similar results were shown by [32], who reported that the degree of crystallinity of sengon wood as a result of MEG and nano-$SiO_2$ impregnation increased from the MEG treatment up to $MEGSiO_2$ 1%. The XRD analysis results showed the effect of adding nano-silica to values for dimensional stability parameters and the resulting oven-dried density. This effect was assisted by the appearance of silica peaks detected in the diffraction plane based on the JCPDS database, and it was concluded that the added nano-silica increased the crystallinity degree.

### 4.1.3. FTIR Analysis

In the wood sample, the functional group detected at wave number 3400 cm$^{-1}$ was a functional group -OH stretching; at wave number 1730 cm$^{-1}$, it was a functional group C=O stretching from the acetyl group; then, at wave number 1261 cm$^{-1}$, it was a functional group CO of guaiasil ring stretching [39]. In sengon wood, a C–H stretching peak of the -CH$_2$ group was detected, which appeared at wave number 2905.34 cm$^{-1}$, which corresponded to the ethylene glycol peak [40]. The presence of the Si–O–Si functional group indicated the presence of silica in jabon and sengon wood. This was in line with [41], who found that the range of wavenumbers was 1050–1115 cm$^{-1}$. This range of wavenumbers is called the Si–O–Si vibration.

## 5. Conclusions

The impregnation of an MEG and nano-silica mixture had a significant effect on the resulting WPG, WU, ASE, BE, and oven-dried density of sengon wood. Changes in the morphology of the treated sengon wood were indicated by the closure of the pits on the vessel walls (SEM analysis results) and the detection of ethylene glycol (FTIR analysis results) and silica (XRD and FTIR analysis results) in the wood samples. Overall, the MNano-Silica 0.75% treatment was the optimal method to increase the dimensional stability of 5-year-old sengon wood.

**Author Contributions:** Conceptualization, I.R.; methodology, I.R. and F.C.D., A.M. and E.P.; software, F.C.D. and E.P.; validation, I.R., A.M., W.D. and D.N.; formal analysis, F.C.D.; investigation, I.R., F.C.D. and E.P.; resources, I.R.; data curation, F.C.D. and E.P.; writing—original draft preparation, F.C.D.; writing—review and editing, I.R., W.D. and D.N. All authors have read and agreed to the published version of the manuscript.

**Funding:** This research was funded by the Ministry of Education, Culture and Research and Technology of Indonesia (grant no. 1/E1/KP.PTNBH/2021 and grant no. 8/E1/KPT/2021).

**Institutional Review Board Statement:** Not applicable.

**Informed Consent Statement:** Not applicable.

**Data Availability Statement:** Not applicable.

**Acknowledgments:** The authors are grateful for the support of the Ministry of Education, Culture and Research and Technology of Indonesia (grant no. 1/E1/KP.PTNBH/2021 and grant no. 8/E1/KPT/2021).

**Conflicts of Interest:** The authors declare no conflict of interest.

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
