# Peer review of "Dimensional Stability of Treated Sengon Wood by Nano-Silica of Betung Bamboo Leaves"

_forests, doi:10.3390/f12111581_

Round 1

Reviewer 1 Report

The work touches on an important topic in this day and age. Unfortunately, the topic presented in the work has not been adequately exhausted in my opinion. 

The abstract lacks the purpose of the work that the paper addresses. 

Sentences from the abstract: "The technology of impregnation can improve the quality of fast-growing wood such as sengon wood. In particular, impregnation with nano silica can improve the dimensional stability and density of sengon wood" . Should go in the literature review. They do not address the subject of what has been done or why it is being done. 

There are 16 literature references, most of which are by the authors of the paper. Please verify the literature items and expand them with other items. 
The abstract is worryingly short. It does not touch the issue of wood impregnation, its types, the purpose of this process (besides dimensional stability of wood, e.g. resistance to the so-called biological corrosion). Other compounds used for wood impregnation (including polymers used e.g. for archaeological wood, WPC wood plastic composites etc.). 

Example articles to complement the literature review: 

DOI 10.1007/s00226-009-0253-6

DOI 10.1007/s00107-015-0998-6

https://doi.org/10.1515/HF.2011.134

https://doi.org/10.1007/s00107-014-0823-7

https://doi.org/10.1007/s10853-011-5480-1

https://doi.org/10.14314/polimery.2019.5.3

DOI: 10.12841/wood.1644-3985.C19.20 

These items are examples and are definitely not exhaustive. The paper lacks an explanation of why wood of fast-growing species is used in industry. 

The analysis of results lacks a test of significance of differences (e.g. ANOVA test).  Which will statistically show whether there are significant differences. In addition, the standard deviation is missing e.g. in the crystallinity index. 

Author Response

Thank you very much for reviewers’ suggestions and corrections to our manuscript.  We have made some changes and revisions according to reviewers’ suggestions.  Our responds to each reviewer suggestions have been made also (represented by sentences in blue colour in this cover letter). Cover letter is attached

Reviewer 2 Report

Please find my comments in the attached file. 

Author Response

(The authors gave the same response as above.)

Round 2

Reviewer 1 Report

The literature review still lacks information on various methods of modifying wood to improve its performance. I understand that the authors have chosen the most advantageous method in their opinion, but in my opinion other possibilities "currently available on the market" should be presented in the literature review (technologies introduced to the industry or being introduced: e.g. densification of the material, modification with plastics in lumen, etc.). Especially since the authors indicate here why the method they have chosen is attractive to the public. 

The literature review states '"Sengon (Falcataria moluccana Miq.) is a type of fast-growing tree that is widely planted in community forests and community plantation forests in Indonesia. Owing to the rapid growth of the tree, the wood has low density, strength, and durability, as well as a high portion of juvenile wood. Based on the research results of [1] 5- to 7-year-old jabon and sengon trees contain of 100% juvenile wood." '. In my opinion there is no information as to why this situation occurs. No news about the increase in demand for wood in different types of industries. And that this trend is worldwide (with information on which fast growing species are most common).

DOI:10.1016/j.proenv.2014.03.040

DOI:10.12841/wood.1644-3985.037.07

DOI:10.35812/CelluloseChemTechnol.2021.55.52 

DOI: 10.12841/wood.1644-3985.037.07

https://doi.org/10.3389/fbioe.2015.00072

Author Response

Thank you very much for reviewers’ suggestions and corrections to our manuscript.  We have made some changes and revisions according to reviewers’ suggestions.  Our responds to each reviewer suggestions have been made also (represented by sentences in blue colour in this letter)
